# BOOSTING REAL-WORLD SUPER-RESOLUTION WITH RAW DATA: A NEW PERSPECTIVE, DATASET AND BASELINE

## ABSTRACT

Real-world image super-resolution (Real SR) aims to generate high-fidelity, detail-rich high-resolution (HR) images from low-resolution (LR) counterparts. Existing Real SR methods primarily focus on processing within the RGB domain. In this paper, we pioneer the use of detail-rich RAW data to complement RGB-only Real SR, specifically by utilizing both LR RGB and RAW inputs to generate superior HR RGB outputs. We argue that key image processing steps in Image Signal Processing, such as denoising and demosaicing, inherently result in the loss of fine details, making RAW data a valuable information source. To validate this, we present RealSR-RAW, a comprehensive dataset comprising 10,000 pairs with LR and HR RGB images, along with corresponding LR RAW data, captured across multiple smartphones under varying focal lengths and diverse scenes. Additionally, we propose a novel, general RAW adapter to efficiently integrate RAW data into existing CNNs, Transformers, and Diffusion-based Real SR models by suppressing the noise contained in RAW and aligning distribution. Extensive experiments demonstrate that incorporating RAW data significantly enhances detail recovery and improves Real SR performance across ten evaluation metrics, including both fidelity and perception-oriented metrics. Our findings open a new direction for the Real SR task, with the dataset and code made available to support future research.

## 1 INTRODUCTION

Real-world image super-resolution (Real SR), a fundamental task in image processing, is designed to enhance the resolution and quality of low-resolution (LR) images (Mou et al., 2022; Liu et al., 2022; Zhou et al., 2020; Wu et al., 2024; Liang et al., 2024; Yang et al., 2023; Chen et al., 2024). Numerous studies have developed specialized CNNs, Transformers, and Diffusion models to learn pixel relationships in LR images and generate high-resolution (HR) images with finer details (Chen et al., 2022; Yu et al., 2024; Liu et al., 2023; Zhang et al., 2023b; Sun et al., 2023; Zhang et al., 2024). However, these approaches primarily focus on the RGB domain. As is well known, SR is an ill-posed problem, making it difficult to recover rich details and high-fidelity results by relying solely on detail-limited LR RGB images (Chen et al., 2023a; Huang et al., 2020; Wang et al., 2021a; Peng et al., 2024a; Luo et al., 2024; Yan et al., 2024; Li et al., 2024), as shown in Figure 1.

During the camera imaging process, photons reflected from physical objects are captured by CMOS or CCD sensors to produce RAW images, which cannot be directly perceived by the human visual system (Blahut, 2010; Prasanna & Rai, 2014). A complex image signal processing (ISP) pipeline, involving a number of operations, is then applied to generate a visually observable RGB image (Pitas, 2000), as illustrated in Figure 2. However, certain modules within the ISP pipeline, such as denoising and demosaicing, inevitably lead to the loss of image details. In Figure 2(b) and Sec. 3, we visualize the residual images of bypassing denoising and before and after the denoising and demosaicing, assessing information loss with feedback from human users and Multimodal Large Language Models (MLLMs). We can observe that both users and MLLMs agree that in the vast majority of scenarios, both denoising and demosaicing in ISP can lead to a loss of detail. This analysis reveals that some fine details are indeed lost during ISP, which exacerbates the challenges of the Real SR task. This raises an important question: *Can the LR RAW images, containing rich and original details information, be utilized to assist Real SR in producing more detail-rich and high-fidelity HR images?*

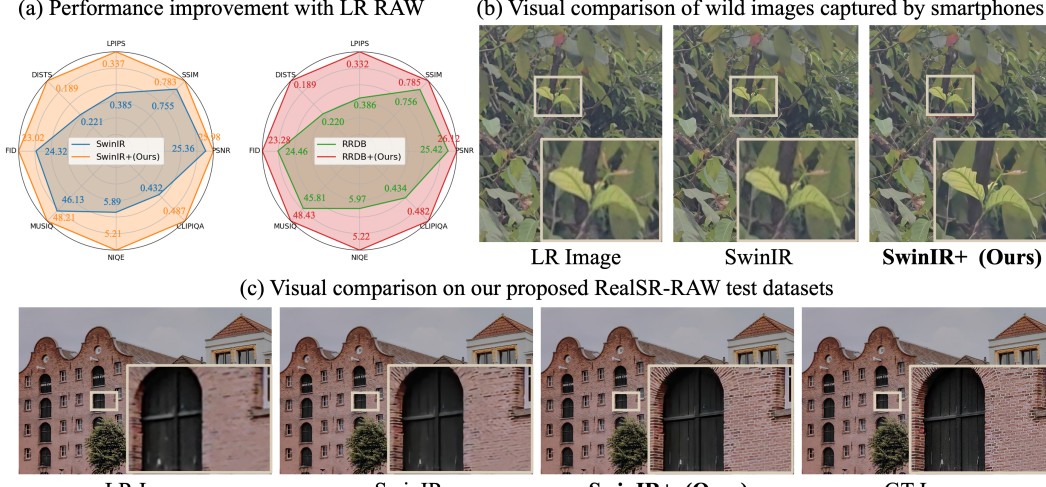

Figure 1: (a) Equipped with LR RAW, the performance of existing RGB-only Real SR models is significantly improved. (b-c) LR RAW also aids Real SR models in generating superior high-fidelity details that are hard to learn in the LR RGB space, thereby significantly enhancing visual quality.

Our answer is **absolutely**. We compare three learning objectives: LR RGB → HR RGB (*i.e.*, Using LR RGB images to generate HR RGB images), LR RAW → HR RGB, and LR RGB + RAW → HR RGB, concluding that the latter, where LR RAW complements LR RGB, delivers the best performance. Since existing Real SR datasets lack paired LR RAW and LR and HR RGB images, we introduce RealSR-RAW, a dataset containing over 10,000 image pairs, including LR RAW and paired LR and HR RGB images. Captured using multiple smartphones across diverse scenes and cameras with different focal lengths, this dataset enables a thorough evaluation of LR RAW's effectiveness. We experiment with three representative Real SR models—CNNs, Transformers, and Diffusion-based methods—and show that simply incorporating LR RAW data largely enhances performance. To maximize the benefits of RAW data, we also design a general RAW adapter to integrate LR RAW information seamlessly into these frameworks by adaptively suppressing noise in LR RAW and aligning the distribution of RAW features to RGB. The results are striking: our approach yields up to 1.109 dB and 0.038 improvements in PSNR and SSIM, consistently producing images with richer, more high-fidelity details, as shown in Figure 1. Our proposed dataset and baseline establish a solid foundation for future research, offering valuable resources for the research community to build upon and further advance the state-of-the-art in Real SR.

The contributions of this paper can be summarized as follows:

- We introduce RealSR-RAW, the first Real SR dataset containing over 10,000 high-quality paired LR and HR RGB images, along with corresponding LR RAW data.
- For the first time, we explore the effectiveness of LR RAW data as a detail supplement to boosting Real SR models, opening a new avenue for advancements in the field.
- To fully leverage LR RAW data, we propose a novel, general RAW adapter that efficiently suppresses noise in RAW and aligns the distribution of RAW features to the RGB domain, resulting in significant improvements across multiple benchmarks and metrics.

## 2    RELATED WORK

**Real-world image super-resolution**. Real-world image super-resolution (Real SR) is an ill-posed problem in image processing, aiming to generate detail-rich and visual pleasing high-resolution images from low-resolution scenes (Lugmayr et al., 2020; Li et al., 2022; Lugmayr et al., 2019; Ji et al., 2020; Mou et al., 2022; Liu et al., 2022; Fritsche et al., 2019b; Zhou et al., 2020; Wang et al., 2021a; Chen et al., 2019). Numerous works have meticulously designed various architecture using CNNs (Wang et al., 2018; 2021a), Transformers (Chen et al., 2023b; Liang et al., 2021b),

Figure 2: Existing RealSR methods focus on LR RGB images, as shown in (a). However, LR RGB images often suffer from detail loss due to ISP, as shown in (b), which exacerbates the challenges of RealSR. Therefore, we think: *Can the detail-rich LR RAW information assist Real SR in generating better image details?*

and Diffusion models (Sun et al., 2023; Yue et al., 2024) to enhance SR performance. For instance, the CNN-based RRDB network (Wang et al., 2018) has been widely adopted in many SR architectures (Wang et al., 2021b; Zhang et al., 2021a; Fritsche et al., 2019a). Liang *et al.* were the first to apply the powerful swin transformer to SR, achieving notable performance (Liang et al., 2021b). Yue *et al.* introduced ResShift, which improves efficiency and performance by generating image residuals through diffusion model (Yue et al., 2024). On the other hand, many researchers also proposed to collect or synthesize paired LR and HR RGB images to enhance the generalization ability of Real SR models in real-world scenarios (Wei et al., 2020; Cai et al., 2019; Peng et al., 2024b; Zhang et al., 2023a). However, existing Real SR methods mainly focus on RGB images with limited details and suffer from addressing this ill-posed problem, thereby leading to over-smooth and low-fidelity details.

**RAW image enhancement**. With the rapid development of smartphone and photography technology, numerous works have focused on enhancing directly to original RAW images (Jiang et al., 2024; Huang et al., 2022; Lu & Jung, 2022; Conde et al., 2022; Heide et al., 2014). For instance, Conde *et al.* organized a RAW SR competition focused on learning the mapping from LR RAW to HR RAW (Conde et al., 2024). Yi *et al.* proposed using diffusion models to establish the mapping from low-quality RAW images captured by smartphones to high-quality RGB images from DSLRs (Yi et al., 2024). Chen *et al.* proposed a model to directly reconstruct normal-exposure RGB images from low-light RAW images (Chen et al., 2018). Xu *et al.* first synthesized LR RAW and RGB images from HR RAW and then proposed to learn color transformation from LR RGB and performed enhancement in the RAW space to generate HR images (Xu et al., 2019). Burst Image Super-Resolution was proposed to generate a high-resolution RGB image directly from a series of LR RAW images captured by burst photography (Bhat et al., 2021). To the best of our knowledge, we are the first to collect real paired images with LR RGB, HR RGB, and LR RAW and explore the benefits of LR RAW as a detail supplement to boost the representation capability of image details for Real SR.

## 3 WHY LR RAW DATA CAN BOOST REAL SR?

### 3.1 IMAGE SIGNAL PROCESSING

In the camera imaging process, photons are captured by CMOS or CCD sensors, which measure light intensity and produce a Bayer RAW image. Since RAW data is in Bayer format and only contains a single color channel per pixel, it cannot be directly interpreted by the human visual system. To

convert RAW data into a perceptually meaningful RGB image, a complex Image Signal Processing (ISP) (Prasanna & Rai, 2014; Blahut, 2010) pipeline is applied. While the exact composition of ISP pipelines can vary significantly across different cameras and devices, certain core operations are universally implemented. For example, demosaicing reconstructs full-color RGB images from the mosaic-like pattern of the Bayer filter, while denoising reduces noise introduced by sensor limitations, high ISO levels, and photon shot noise. Color correction is employed to map the device-specific sensor response to a standardized color space, while white balance adjustment further refines this process by compensating for lighting conditions, and neutralizing color casts caused by the ambient light's color temperature. Another essential operation is defective pixel correction, which addresses sensor irregularities by identifying and interpolating faulty pixel data to maintain image consistency. Collectively, these steps play a pivotal role in converting sensor data into high-quality RGB images.

However, certain processes within the ISP pipeline, such as denoising (Tian et al., 2020; Fan et al., 2019) and demosaicing (Li et al., 2008; Li, 2005), inevitably result in the loss of fine details in the final RGB image, as discussed in Section 3.2. This loss poses significant challenges for Real SR methods operating solely in the RGB domain, making it difficult to reconstruct detail-rich and high-fidelity HR images from the degraded LR RGB data.

## 3.2 DETAIL LOSS DURING IMAGE SIGNAL PROCESSING

To avoid copyright concerns related to commercial ISPs, we analyze the problem of detail loss using an open-source available ISP, OpenISP[1], and the widely-used RAW processing library, RawPy[2], on the MIT-Adobe FiveK dataset (Bychkovsky et al., 2011). Specifically, we employ two analysis methods: bypassing and step-by-step analysis to explore individual ISP modules and compare the resulting images, as well as to analyze the image differences before and after processing through specific modules. Given that image details are mainly characterized as high-frequency signals, we focus on modules that potentially impact high-frequency information, such as the denoising and demosaicing process, in order to investigate detail loss during the ISP. More analyses of other modules in ISP are presented in Appendix A.2. For a comprehensive evaluation, we involve both human volunteers and Multimodal Large Language Models (MLLMs) to assess information loss.

**Bypass analysis.** Using the RawPy library, we process RAW data both bypassing and non-bypassing the denoising module to generate two RGB images for comparison. As shown in Figure 2, the image bypassing denoising exhibits a certain level of noise but retains more image details. We further compute the residual between the two images, revealing more structural details alongside some noise, as visualized in Figure 2(b) and Figure 7 in the appendix. This suggests that detail loss occurs during denoising. To verify this systematically, we randomly select 100 RAW images from the FiveK dataset and repeat the bypass/non-bypass operations. We present the paired RGB images and residuals to ten volunteers, asking: *{USER: Please determine if the residual image on the right contains the structural content information of the image on the left? Answer Yes or No.}* Additionally, we utilize the MLLM model LLava1.5 (Liu et al., 2024) to evaluate a larger test set consisting of 1000 images. The results indicate that in **98%** of the scenarios, ten volunteers agree that the residuals contain detailed structural information, with LLava corroborating this in **95.4%** of the cases.

**Step-by-step analysis.** We also perform a step-by-step analysis of the denoising and demosaicing processes using OpenISP to explore potential detail loss. Specifically, we visualize the images before and after the denoising and demosaicing and then visualize the residual images to analyze the difference introduced during this process. As shown in Figure 8 and 9 in Appendix, the results show that these residuals retain substantial structural information. To further assess this, ten volunteers and MLLMs are invited for evaluation as above analysis. The results indicate that in **99%** and **98%** of the scenarios, volunteers recognize detailed structural information in the denoising and demosaicing residuals, respectively, while LLava reaches the same conclusion in **98.2%** and **97.9%** of the scenes.

From the above analysis, it is clear that detail loss occurs throughout the ISP pipeline. As a result, performing Real SR in the LR RGB domain poses significant challenges in recovering detail-rich and high-fidelity images due to the ill-posed nature of the task. To address this, we propose leveraging LR RAW to enhance Real SR and achieve better reconstruction of finer image details.

---

[1]https://github.com/cruxopen/openISP

[2]https://pypi.org/project/rawpy

Table 1: Comparison with existing Real SR data, where "w/" indicates "with". For the first time, we collect over 10,000 scenes with paired LR RAW, LR RGB, and HR RGB.

| Dataset | w/ HR | w/ LR | w/ RAW | Number |
|---------|-------|-------|--------|--------|
| DIV2K | ✓ | ✗ | ✗ | 800 |
| UHD4K | ✓ | ✗ | ✗ | 8,099 |
| RealSR | ✓ | ✓ | ✗ | 559 |
| DRealSR | ✓ | ✓ | ✗ | 2,000 |
| **Ours** | ✓ | ✓ | ✓ | **11,726** |

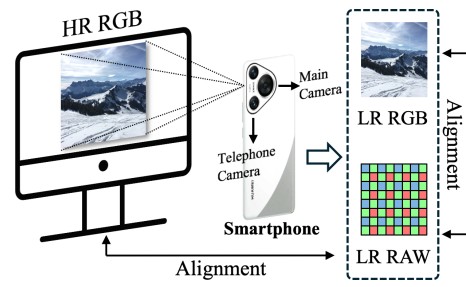

Figure 3: Illustration of data collection and the following alignment methods.

## 4 DATASET AND METHOD

### 4.1 REALSR-RAW DATASET

As shown in Table 1, current real-world super-resolution datasets, such as DIV2K (Agustsson & Timofte, 2017), UHD4K (Zhang et al., 2021b), RealSR (Cai et al., 2019), and DRealSR (Wei et al., 2020), are limited to RGB images and offer a relatively small number of paired samples, which hampers their diversity and broader applicability.

To unlock the potential of LR RAW data, we present RealSR-RAW, the first large-scale dataset comprising over 10,000 diverse scenes with paired LR RAW, LR RGB, and HR RGB images. Specifically, we first gather high-quality 4K+ resolution images from the open-source platform Unsplash[3]. To ensure compliance, we contact Unsplash's official team to receive support and remove any images with potential copyright or ethical concerns. These high-quality images are then displayed on ultra high-definition monitors, where we capture LR RGB and LR RAW images using the main and telephoto cameras of HUAWEI Mate 50 Pro and P70 phones at different focal lengths. The original high-resolution images are used as ground truth HR images. Finally, we apply a two-stage alignment process: first aligning the LR RGB images to their corresponding LR RAW counterparts, and then aligning the HR RGB images to the LR data using estimated homography matrices and optical flow, as shown in Figure 3. We also perform color correction to ensure color-consistent pairs. In total, we collect 11,726 image pairs, which are divided into a training subset and a test benchmark. The resolution of the LR RGB and RAW images ranges from approximately 1K to 2K, while the HR RGB images range from 2K to 4K, with a scaling factor of 2. More details are provided in Appendix A.3. Our dataset will be made open-source to facilitate community research.

### 4.2 REAL SR WITH LR RAW CONCATENATION

Popular Real SR methods reconstruct a high-quality HR image, $\mathcal{HR}_{RGB}$, from a LR RGB image, $\mathcal{LR}_{RGB}$, using a dedicated SR model. The SR model typically consists of a shallow feature extraction module, $L_s$, a feature enhancement module, $L_e$, and a feature-to-image mapping layer, $L_f$:

$$\mathcal{HR}_{RGB} = L_f\left(L_e\left(L_s\left(\mathcal{LR}_{RGB}\right)\right)\right). \tag{1}$$

Building on this formulation, a straightforward strategy to introduce RAW images is to concatenate the LR RAW image with the LR RGB image $\mathcal{LR}_{RAW}$ as the input, which can be expressed as:

$$\mathcal{SR}_{RGB} = L_f\left(L_e\left(L_s\left(\mathcal{LR}_{RGB}\|\mathcal{LR}_{RAW}\right)\right)\right). \tag{2}$$

where $\|$ is the concatenation operation. We are surprised to find that this simple approach largely improves the performance of the Real SR model, as demonstrated in Section 5.2, highlighting the effectiveness of incorporating RAW data.

### 4.3 REAL SR WITH RAW ADAPTER

Considering that LR RAW is in Bayer format and contains an amount of noise, directly concatenating LR RAW and RGB images can lead to distribution mismatches and noise interference. To address

---

[3]https://unsplash.com/

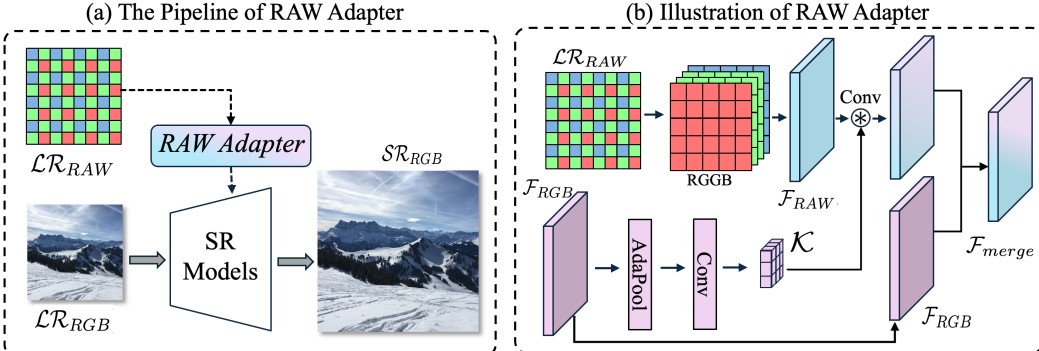

Figure 4: (a) The proposed RAW adapter seamlessly integrates into various popular Real SR methods to boost their representation capability of detail. (b) Illustration of the proposed RAW adapter.

these issues, we propose a general and efficient RAW adapter that facilitates the fusion of LR RGB and RAW in the feature space, fully leveraging the potential of RAW information. Also, this adapter can be seamlessly integrated into various Real SR models, as illustrated on the left of Figure 4.

In detail, as shown on the right of Figure 4, we first use shallow feature extractors $L_s^{RGB}$ and $L_s^{RAW}$ to process the LR RGB and LR RAW, producing $\mathcal{F}_{RGB}$ and $\mathcal{F}_{RAW}$. Specifically, $L_s^{RAW}$ unpacks the Bayer format into RGGB channels, applies convolutional blocks to extract features, and utilizes transposed convolution to upsample the resolution, matching it with the RGB features. Next, adaptive kernels are generated from $\mathcal{F}_{RGB}$ to convolve with $\mathcal{F}_{RAW}$, producing $\mathcal{F}_{RAW}^{'}$, which is aligned with the RGB features. These adaptive kernels, $\mathcal{K}$, are obtained by performing adaptive pooling and convolution on $\mathcal{F}_{RGB}$ to perceive the distribution of RGB while modulating learnable kernels, $\mathcal{K}_{learn}$. Finally, we concatenate $\mathcal{F}_{RAW}^{'}$ with $\mathcal{F}_{RGB}$, followed by a convolution to produce the fused result, $\mathcal{F}_{merge}$, and carry out the reconstruction HR image $\mathcal{SR}_{RGB}$. This process is expressed as:

$$\mathcal{F}_{RGB} = L_s^{RGB}\left(\mathcal{LR}_{RGB}\right), \quad \mathcal{F}_{RAW} = L_s^{RAW}\left(\mathcal{LR}_{RAW}\right), \quad (3)$$

$$\mathcal{K} = Conv\left(AdaPool\left(\mathcal{F}_{RGB}\right)\right) \cdot \mathcal{K}_{learn}, \quad \mathcal{F}_{RAW}^{'} = \mathcal{F}_{RAW} * \mathcal{K}, \quad (4)$$

$$\mathcal{F}_{merge} = Conv\left(\mathcal{F}_{RAW}^{'} \| \mathcal{F}_{RGB}\right), \quad \mathcal{SR}_{RGB} = L_f\left(L_e\left(\mathcal{F}_{merge}\right)\right). \quad (5)$$

This design offers two key advantages. First, it adaptively fuses RAW features based on the distribution of individual RGB images, greatly enhancing model flexibility. Second, using noise-free RGB features to generate the kernels improves the extraction of useful details from RAW data while mitigating the influence of noise. As demonstrated in Table 6, the proposed RAW adapter significantly elevates model performance compared with the simple concatenation.

## 5 EXPERIMENTS AND ANALYSIS

### 5.1 IMPLEMENTATION

**Training Details.** To evaluate the impact of LR RAW data, we compare the traditional RGB-only LR RGB input with our proposed RAW adapter using LR RGB + LR RAW input for Real SR under consistent experimental settings. All experiments are conducted at the ×2 super-resolution scale using the L1 loss function for training and evaluation unless otherwise specified. Further training details and elevation on perceptual-oriented GAN loss are provided in Section 5.2 and Appendix A.4.

**Real SR models.** We conduct experiments on three popular and representative Real SR models, including the CNN-based RRDB network (RRDB) (Wang et al., 2018; 2021b), the transformer-based model SwinIR (Liang et al., 2021a), and the diffusion-based model ResShift (Yue et al., 2024).

**Metrics.** To comprehensively evaluate the quality of generated images, we employ a total of ten widely-used and popular image quality assessment metrics for evaluation, including four reference-based metrics: PSNR↑ (Huynh-Thu & Ghanbari, 2008), SSIM↑ (Wang et al., 2004), LPIPS↓ (Zhang

Table 2: Performance comparison on SwinIR and RRDB models. The model and model+ represent the Real SR model with traditional mapping LR RGB → HR RGB, and our proposed RAW adapter (LR RGB + RAW → HR RGB), respectively. "M50" refers to the Mate 50 Pro phone, while "P70" denotes the Pura 70 phone. "M" and "T" indicate the main and telephoto cameras, respectively.

| Dataset | Models | PSNR↑ | SSIM↑ | LPIPS↓ | DISTS↓ | FID↓ | MUSIQ↑ | NIQE↓ | CLIP-IQA↑ |
|---------|--------|-------|-------|--------|--------|------|--------|-------|-----------|
| M50-M | SwinIR | 25.367 | 0.755 | 0.385 | 0.221 | 24.326 | 46.137 | 5.890 | 0.432 |
| | SwinIR+ | 25.982 | 0.783 | 0.337 | 0.189 | 23.029 | 48.217 | 5.216 | 0.487 |
| | Gain | 0.615 | 0.028 | 0.048 | 0.032 | 1.297 | 2.080 | 0.674 | 0.055 |
| | RRDB | 25.426 | 0.756 | 0.386 | 0.220 | 24.462 | 45.811 | 5.979 | 0.434 |
| | RRDB+ | 26.126 | 0.785 | 0.332 | 0.189 | 23.287 | 48.437 | 5.227 | 0.482 |
| | Gain | 0.700 | 0.029 | 0.054 | 0.031 | 1.175 | 2.626 | 0.752 | 0.048 |
| M50-T | SwinIR | 25.181 | 0.749 | 0.322 | 0.206 | 3.976 | 39.634 | 6.156 | 0.430 |
| | SwinIR+ | 25.588 | 0.766 | 0.296 | 0.185 | 3.239 | 40.143 | 5.815 | 0.455 |
| | Gain | 0.407 | 0.017 | 0.026 | 0.021 | 0.737 | 0.509 | 0.341 | 0.025 |
| | RRDB | 25.279 | 0.753 | 0.316 | 0.203 | 4.032 | 40.047 | 6.022 | 0.447 |
| | RRDB+ | 25.720 | 0.770 | 0.290 | 0.182 | 3.192 | 40.665 | 5.764 | 0.469 |
| | Gain | 0.441 | 0.017 | 0.026 | 0.021 | 0.840 | 0.618 | 0.258 | 0.022 |
| P70-M | SwinIR | 24.642 | 0.776 | 0.300 | 0.173 | 3.263 | 46.745 | 4.646 | 0.508 |
| | SwinIR+ | 25.744 | 0.815 | 0.251 | 0.145 | 2.391 | 49.076 | 4.339 | 0.539 |
| | Gain | 1.102 | 0.039 | 0.049 | 0.028 | 0.872 | 2.331 | 0.307 | 0.031 |
| | RRDB | 24.836 | 0.781 | 0.295 | 0.169 | 3.221 | 46.886 | 4.630 | 0.520 |
| | RRDB+ | 25.945 | 0.819 | 0.242 | 0.142 | 2.387 | 49.406 | 4.321 | 0.556 |
| | Gain | 1.109 | 0.038 | 0.053 | 0.027 | 0.834 | 2.520 | 0.309 | 0.036 |
| P70-T | SwinIR | 24.753 | 0.735 | 0.356 | 0.220 | 8.077 | 38.593 | 6.305 | 0.417 |
| | SwinIR+ | 25.108 | 0.749 | 0.334 | 0.203 | 5.603 | 39.412 | 6.056 | 0.431 |
| | Gain | 0.355 | 0.014 | 0.022 | 0.017 | 2.474 | 0.819 | 0.249 | 0.014 |
| | RRDB | 24.829 | 0.737 | 0.354 | 0.220 | 7.874 | 39.224 | 6.269 | 0.426 |
| | RRDB+ | 25.185 | 0.751 | 0.332 | 0.204 | 5.605 | 39.917 | 6.029 | 0.437 |
| | Gain | 0.356 | 0.014 | 0.022 | 0.016 | 2.269 | 0.693 | 0.240 | 0.011 |

Table 3: Performance comparison of different learning mappings for ResShift on the M50-M dataset.

| Methods | PSNR↑ | SSIM↑ | LPIPS↓ | DISTS↓ | FID↓ | MUSIQ↑ | CLIP-IQA↑ |
|---------|-------|-------|--------|--------|------|--------|-----------|
| ResShift | 24.809 | 0.732 | 0.330 | 0.173 | 24.310 | 47.528 | 0.447 |
| **ResShift+** | **25.071** | **0.761** | **0.312** | **0.161** | **23.790** | **47.882** | **0.449** |

et al., 2018), and DISTS↓ (Ding et al., 2020); and six no-reference metrics: FID↓ (Heusel et al., 2017), MUSIQ↑ (Ke et al., 2021), NIQE↓ (Mittal et al., 2012), CLIP-IQA↑ (Radford et al., 2021), NIMA↑ (Talebi & Milanfar, 2018), and MANIQA↑ (Yang et al., 2022). Note that ↑ and ↓ indicate that higher and lower values respectively represent better image quality.

## 5.2 QUANTITATIVE AND QUALITATIVE RESULTS

**RealSR-RAW benchmark.** To demonstrate the improvements that RAW images can bring to Real SR, we utilize three popular Real SR models and compare the different learning mappings: LR RGB → HR RGB, and our proposed RAW adapter (LR RGB + RAW → HR RGB). Note that since the official implementation of ResShift only supports the super-resolution factor of × 4, its performance is also evaluated at this scale. As shown in Table 2 and 3, our method largely surpasses traditional LR RGB → HR RGB approach across all benchmarks and metrics. For instance, compared to traditional Real SR, our method improves PSNR by 1.109 dB and LPIPS by 0.053 for the RRDB model on the P70-M dataset. Furthermore, we are surprised that simply inserting the RAW image into model input also achieves considerable gains, as shown in Table 6, demonstrating the significant potential of RAW images for Real SR. As shown in Figure 1 and 5, we can observe that, compared to LR RGB → HR RGB, our proposed method assists the RealSR model in extracting more image details from RAW data, generating higher fidelity and detail-rich HR images.

Table 4: Performance comparison on in-the-wild LR images captured by Mate 50 Pro.

| Devices | Models | MUSIQ↑ | NIQE↓ | CLIP-IQA↑ | NIMA↑ | MANIQA↑ |
|---------|--------|--------|-------|-----------|-------|---------|
| Main Camera | RRDB | 31.940 | 7.867 | 0.300 | 3.687 | 0.253 |
| | **RRDB+** | **34.313** | **7.850** | **0.310** | **3.767** | **0.260** |
| | SwinIR | 32.088 | 7.815 | 0.299 | 3.708 | 0.252 |
| | **SwinIR+** | **35.060** | **7.677** | **0.309** | **3.842** | **0.262** |
| Telephone Camera | RRDB | 47.313 | 7.346 | 0.431 | 4.181 | 0.284 |
| | **RRDB+** | **48.406** | **7.314** | **0.445** | **4.359** | **0.290** |
| | SwinIR | 45.965 | 7.532 | 0.414 | 4.250 | 0.275 |
| | **SwinIR+** | **46.562** | **7.480** | **0.420** | **4.305** | **0.282** |

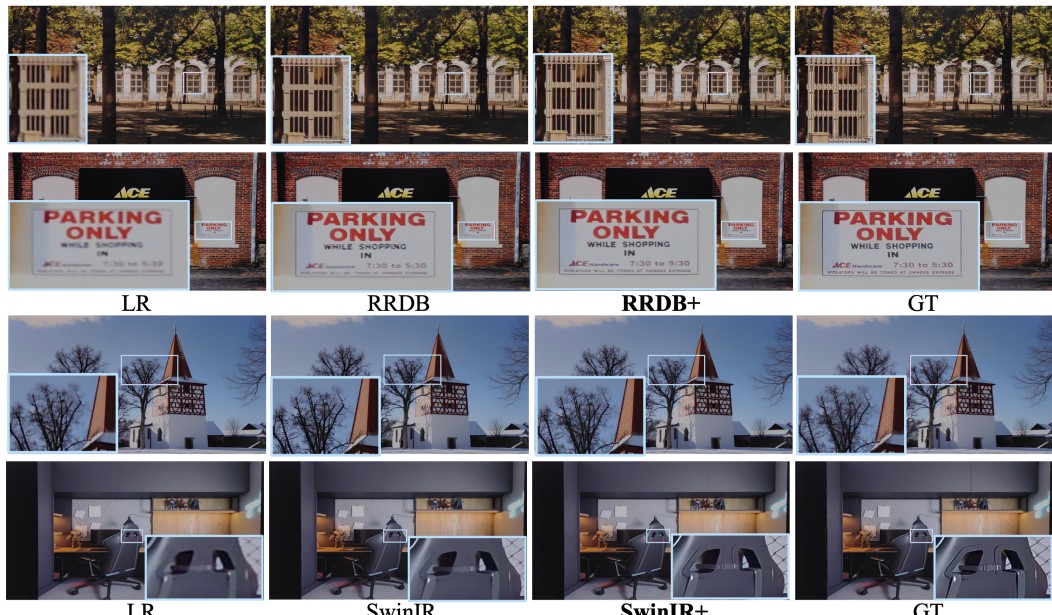

Figure 5: Visual comparison of RRDB and SwinIR on our RealSR-RAW dataset.

**Real-world test images.** To evaluate the effectiveness of RAW data in real-world scenarios, we use the main and telephoto cameras of the Mata 50 Pro to capture 192 and 224 pairs of LR RGB and LR RAW images, respectively. RRDB models pre-trained on Mata 50 Pro datasets are applied for evaluation. Since real-world test images in the wild lack ground truth, we employ five widely used no-reference metrics for assessment. As shown in Table 4, incorporating LR RAW significantly improves the Real SR model's performance across all metrics. Figure 1(c) and Figure 12 in the appendix illustrate that our method is also capable of generating HR images with richer textures.

**User study.** We conduct a user study using 10 randomly selected real LR images captured by the Mata 50 Pro, evaluating the performance of RRDB and SwinIR. Ten volunteers are invited to rate the quality of the generated images on a scale of 1 to 10. As shown in Figure 6, RRDB+ and SwinIR+, enhanced by our RAW adapter, achieve higher average scores of 7.92 and 7.99, respectively, due to improved detail representation from RAW data. These results demonstrate the effectiveness of our approach in enhancing visual quality.

**Validation of GAN loss.** We also compare Real SR model performance on P70-M when trained using the commonly

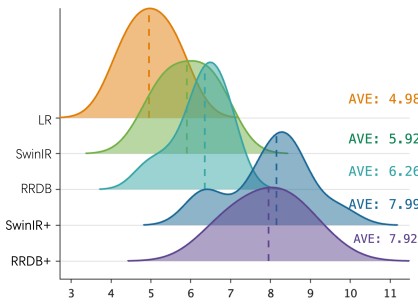

Figure 6: User study of real images on RRDB and SwinIR models.

Table 5: Performance comparison of RRDB under different mappings, trained using GAN loss.

| Models | PSNR↑ | SSIM↑ | LPIPS↓ | DISTS↓ | FID↓ | MUSIQ↑ | NIQE↓ | CLIP-IQA↑ |
|--------|-------|-------|--------|--------|------|--------|-------|-----------|
| RRDB | 23.002 | 0.723 | 0.188 | 0.102 | 9.810 | 57.376 | 3.315 | 0.670 |
| **RRDB+** | **24.053** | **0.766** | **0.158** | **0.087** | **8.921** | **57.868** | **3.301** | **0.673** |

Table 6: Performance and computational complexity of RRDB with different mapping. The input size is $3\times224\times224$. LR RGB + RAW →HR represent our RAW adapter.

| Mapping | Param(M) | FLOPs(G) | Time(s) | PSNR↑ | SSIM↑ | LPIPS↓ |
|---------|----------|----------|---------|-------|-------|--------|
| LR RGB→HR | 9.57 | 482.98 | 0.0795 | 25.426 | 0.756 | 0.386 |
| LR RGB ‖ LR RAW →HR | 9.58 | 482.99 | 0.0796 | 25.913 | 0.779 | 0.339 |
| LR RGB + LR RAW →HR | 9.64 | 485.67 | 0.0799 | **26.126** | **0.785** | **0.332** |

adopted perceptual-oriented GAN loss. Following Wang et al. (2021b), the total loss function is combined with L1, GAN, and VGG loss function. As shown in Table 5, the RAW adapter still consistently enhances performance across all image quality metrics under perceptual-oriented GAN loss. For example, there is a 1.051 dB and 0.03 improvement in the PSNR and LPIPS metric, confirming that the RAW adapter improves the model's ability to perceive finer details.

**Model complexity.** To demonstrate the efficiency of LR RAW and our RAW adapter, we compare model parameters, FLOPs, and inference time with different mappings. As shown in Table 6, it is evident that our method achieves noticeable performance improvements with minimal additional computational overhead. Compared to only using LR RGB and directly concatenating LR RAW images, our RAW adapter is capable of better extracting detailed information from RAW images and integrating it into the RGB feature space, with only a slight increase in computational complexity.

## 5.3 DISCUSSION AND ANALYSIS

**Why not LR RAW → HR RGB?** Considering that LR RAW contains rich information, an intuitive approach might be to directly map LR RAW to HR RGB, using an SR model to generate high-quality HR RGB images from a single LR RAW input. However, generating RGB images from RAW data typically requires complex image processing operations within the ISP, making it difficult for a single SR model to handle the entire LR RAW → HR RGB mapping. For example, HR RGB images adhere to a well-defined color space, which is challenging for a model to reproduce without explicit color correction and adjustment. To validate this, we conduct experiments with RRDB on three different mappings on the P70: $\mathcal{M}1$, $\mathcal{M}2$, and $\mathcal{M}3$. As expected, the LR RAW → HR RGB results show color shifts due to the lack of color adjustment, leading to lower performance compared to traditional Real SR methods like LR RGB → HR RGB, as shown in Table 7. In contrast, our proposed RAW adapter effectively extracts detailed information from RAW images to enhance RealSR performance in the RGB space. Additionally, the lack of large-scale, high-quality LR RAW → HR RGB datasets may have hindered

Table 7: Performance comparison of the RRDB backbone across different mappings. $\mathcal{M}1$: LR RGB → HR RGB, $\mathcal{M}2$: LR RAW → HR RGB, and $\mathcal{M}3$: LR RGB+RAW → HR RGB.

| Mapping | P70-M | | P70-T | |
|---------|-------|-------|-------|-------|
| | PSNR↑ | SSIM↑ | PSNR↑ | SSIM↑ |
| $\mathcal{M}1$ | 24.833 | 0.777 | 24.826 | 0.734 |
| $\mathcal{M}2$ | 22.938 | 0.739 | 23.205 | 0.709 |
| $\mathcal{M}3$ | **25.960** | **0.819** | **25.192** | **0.751** |

further exploration in this area. Nonetheless, this approach holds significant research and practical potential, which we plan to investigate in the future.

**Why not LR RAW → HR RAW → ISP?** We identify three main challenges with this pipeline: (a) It is difficult for a single SR model to learn clean mappings from noisy LR RAW inputs. As a result, any remaining noise or artifacts introduced by the SR model will be amplified during ISP, leading to degraded image quality. (b) The RAW space lacks sufficient image/model priors, such as those

Table 8: Cross-lens generalization performance. The M50→P70 indicates the generalization performance on the main camera of P70 test data using the pre-trained model of the M50. It can be observed that the proposed RAW adapter, LR RGB+RAW → HR RGB, still significantly outperforms LR RGB → HR RGB in cross-lens scenarios.

| Cross-Lens | Model | PSNR↑ | SSIM↑ | LPIPS↓ | DISTS↓ | FID↓ | MUSIQ↑ | NIQE↓ | CLIP-IQA↑ |
|---|---|---|---|---|---|---|---|---|---|
| M50→P70 | RRDB | 22.654 | 0.685 | 0.450 | 0.213 | 8.525 | 30.892 | 6.236 | 0.531 |
|  | **RRDB+** | **22.825** | **0.705** | **0.419** | **0.196** | **7.439** | **35.123** | **5.477** | **0.534** |
| P70→M50 | RRDB | 24.512 | 0.721 | 0.466 | 0.261 | 28.611 | 37.724 | 7.001 | 0.363 |
|  | **RRDB+** | **25.203** | **0.749** | **0.360** | **0.206** | **26.069** | **52.395** | **5.014** | **0.518** |

available in the Stable Diffusion models (Rombach et al., 2022), making it harder to design a powerful RAW-based model. In contrast, the RGB space can leverage these priors for better reconstruction, which is a key insight behind our approach that combines the strengths of both RGB and RAW spaces. (c) Increasing image resolution in the RAW space before ISP significantly raises the computational load of ISP, making this pipeline impractical for edge devices like smartphones and cameras.

**Improvement gaps between main and telephoto cameras.** As shown in Tables 2, different SR models exhibit varying performance improvements between the main and telephoto cameras on P70 and Mate 50 Pro phones. For instance, in the RRDB model with our proposed RAW adapter, the PSNR improvement on the P70's main camera is 1.109 dB, while it is only 0.356 dB on the telephoto camera. These discrepancies may arise from differences in sensor quality. Manufacturers typically prioritize enhancing the main camera, as it is the most frequently used, resulting in a higher-quality sensor that captures more detailed information in RAW images. Consequently, our RAW adapter is more effective at extracting detail from the main camera, leading to greater performance gains in Real SR models.

**Cross-Lens generalization ability.** We conduct cross-lens experiments to evaluate the generalization capability of the RAW adapter under different lens conditions. Specifically, we perform cross-lens tests between the main cameras of the M50 and P70 smartphones, using RRDB models pretrained on the M50 and P70 to evaluate the test sets of P70 and M50, respectively. As shown in Table 8, the RAW adapter consistently improves performance in both M50→P70 and P70→M50 scenarios. For example, in the P70→M50 test, our RAW adapter boosts PSNR by 0.691 dB, SSIM by 0.028, and LPIPS by 0.106. These results demonstrate that the proposed RAW adapter exhibits strong generalization across different lenses.

To thoroughly examine the effectiveness of our proposed RAW adapter, we provide additional discussions, in-depth analysis, and extensive visual comparisons in the Appendix. These supplementary materials offer further insights into performance improvements across various scenarios, and highlight the adapter's robustness.

## 6 CONCLUSION

In this paper, we explore the potential of leveraging LR RAW data as a detailed supplement to enhance real-world super-resolution, overcoming the limitations of traditional RGB-only Real SR methods. We also introduce the RealSR-RAW dataset for community research, consisting of over 10,000 high-quality paired images, including LR RGB, HR RGB, and LR RAW data. Furthermore, we propose a novel RAW adapter that adaptively suppresses noise in RAW data and aligns RAW features with the RGB domain, improving the detail recovery of various existing Real SR models and producing high-fidelity, detail-rich HR images. Extensive experiments demonstrate that our RAW adapter significantly enhances the visual quality of current Real SR methods across all metrics. We hope that our dataset and findings will open new avenues for Real SR research.

In the future, we aim to design more advanced SR models to fully harness the detailed information in RAW data and integrate it with RGB for improving Real SR and other low-level vision tasks. Additionally, we plan to expand our datasets by collecting more RAW data from a wider range of devices, enhancing both the quality and quantity of the data. Furthermore, since other metadata within the ISP is available during camera deployment, we believe that utilizing this information alongside RAW images presents a promising opportunity for further improving the quality of generated images.

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

# A  APPENDIX

## A.1  MORE VISUAL RESIDUAL ANALYSIS

**Bypass analysis.** Here, we provide more residual analysis visualizations, as shown in Figure 7. It can be seen that the residuals on the right of all scenes contain most of the image detail information, leading us to conclude that denoising can lead to a loss of detail.

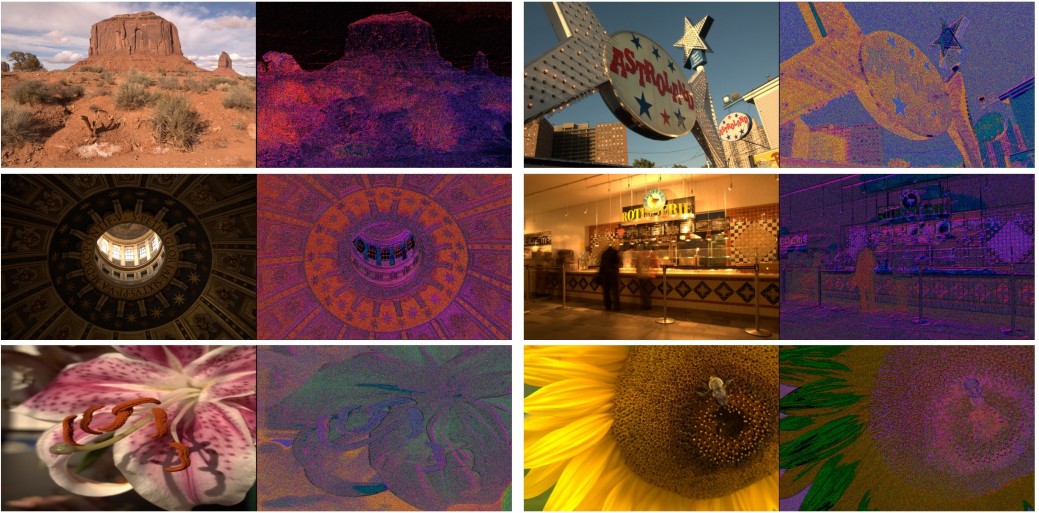

Figure 7: Visualizations of bypass analysis method. On the right of each scene is the denoised sRGB image, and on the left are the residuals with and without bypass denoising.

**Step-by-step analysis.** Here, we provide more residual analysis visualizations of denoising and demosaicing, as shown in Figure 8 and 9. It can be seen that the residuals on the right of all scenes contain most of the image detail information, leading us to conclude that denoising and demosaicing can lead to a loss of detail.

## A.2  MORE ANALYSES OF OTHER MODULES IN ISP.

**Analyses of Image Stabilization.** In Section 3, we demonstrate that the denoising and demosaicing modules within the ISP can degrade image details. Additionally, considering that modern smartphones are often equipped with image stabilization systems, utilizing an internal gyroscope to estimate a motion trajectory warp matrix in the YUV space and apply it to images, we further investigate whether this process results in detail loss by simulating both a warp and an unwarp matrix. We select 100 images from the Mate 50 Pro test set for processing and analyze the original and warped-unwarped

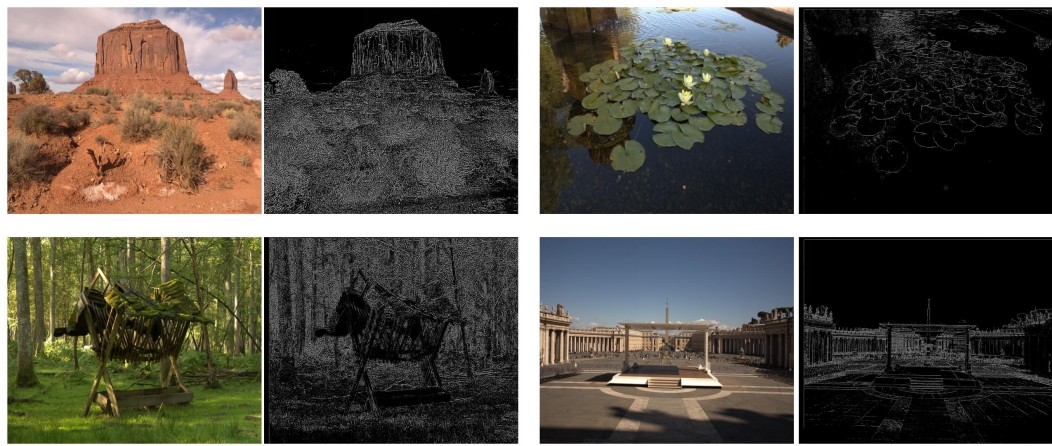

Figure 8: Visualizations of step-by-step analysis method. On the right of each scene is the denoised sRGB image, and on the left are the residuals after and before denoising.

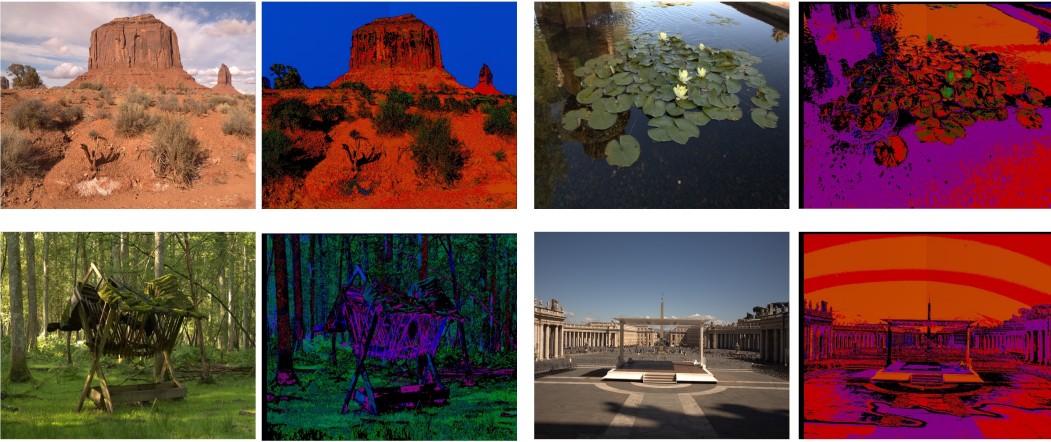

Figure 9: Visualizations of step-by-step analysis method. On the right of each scene is the denoised sRGB image, and on the left are the residuals after and before demosaicing.

images. By subtracting these, we derive residual images for analysis and visualization, as shown in Figure 10 and 11.

Ten volunteers participate in an evaluation, being asked: *USER: Please determine if the residual image on the right contains the structural content information of the image on the left. Answer Yes or No.* The results indicate that in 97% of the scenarios, volunteers agree that the residuals contain detailed structural information.

A.3 DETAIL OF OUR COLLECTED DATASET.

Table 9: Details of our collected data. Number of training and testing samples used in this study.

| Smartphone | Mate 50 Pro | | P70 | | Total |
| | Main Camera | Telephoto Camera | Main Camera | Telephoto Camera | |
|---|---|---|---|---|---|
| Train | 2,600 | 2,800 | 2,694 | 2,800 | 10,894 |
| Test | 220 | 202 | 218 | 192 | 832 |

In this section, we provide a detailed overview of the collected datasets, including specific quantities of training and testing data. As shown in Table 9, we collected a total of 11,726 paired samples. The

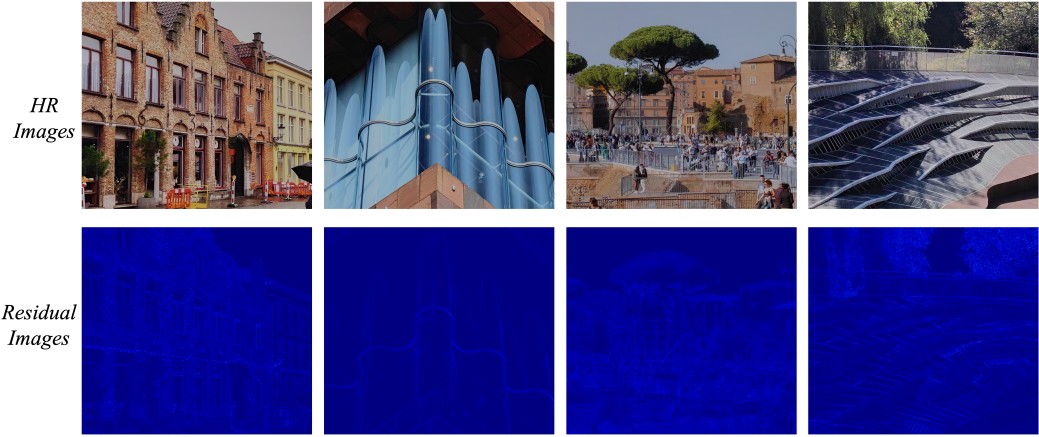

Figure 10: This visualization illustrates the loss of detail introduced by Image Stabilization. Examination of the residual images reveals that a significant amount of image detail is retained.

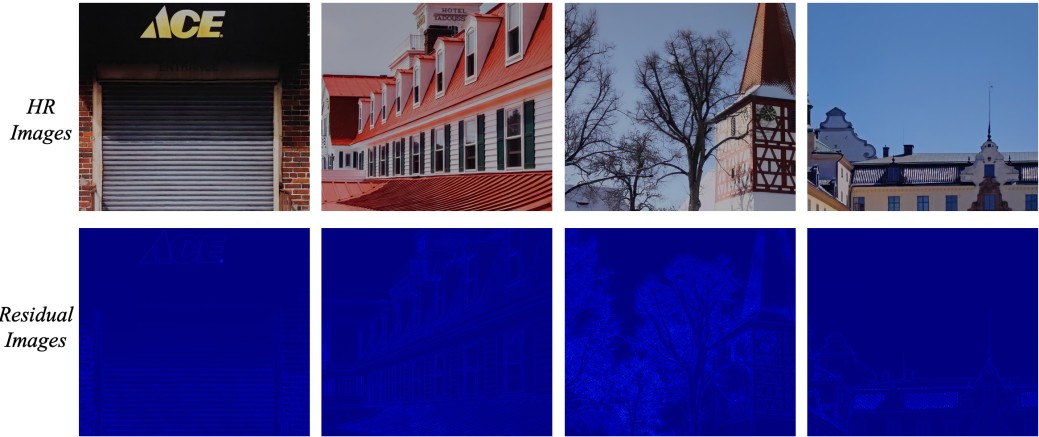

Figure 11: This visualization illustrates the loss of detail introduced by Image Stabilization. Examination of the residual images reveals that a significant amount of image detail is retained.

training and testing data for different focal lengths using Mate 50 Pro and P70 are also detailed in Table 9.

### A.4 MORE TRAINING DETAILS

We use the open-source and widely-used BasicSR framework to conduct experiments on three representative RealSR methods: RRDB, SwinIR, and ResShift. We utilize the public BasicSR for training and evaluate Real-SR methods with a total of 16 NVIDIA V100 GPUs. The training details for each method are as follows:

**Training details of RRDB.** During the training of RRDB, the input size is set to $128 \times 128$, with the resolution of the ground truth set to $256 \times 256$. The batch size per GPU is 4, and we utilize a total of 4 V100 GPUs for training RRDB.

**Training details of SwinIR.** For SwinIR, to facilitate rapid validation, we select the Small version of SwinIR, keeping the resolution of the input image and ground truth consistent with SwinIR. The batch size per GPU is 4, and we use a total of 8 V100 GPUs for training SwinIR.

**Training details of ResShift.** For ResShift, we maintain the settings consistent with the official release. While training ResShift, since our collected dataset is for $2\times$ super-resolution and the official

open-source ResShift only supports 4× super-resolution, we enlarge the HR images by two times to construct a 4× super-resolution for training.

Our proposed RAW adapter aims to extract detailed information from RAW images to assist Real SR, thus we maintain the training scenarios of Real SR completely consistent, with only the mapping differing: one is the traditional Real SR, LR RGB → HR RGB, and our method is LR RGB+RAW → HR RGB.

## A.5 MORE VISUAL COMPARISON.

Here, we present more visual comparisons of real images captured by smartphones in the wild, as shown in Figure 12. It can be observed that our proposed method achieves richer detail and superior visual quality in real-world scenarios.

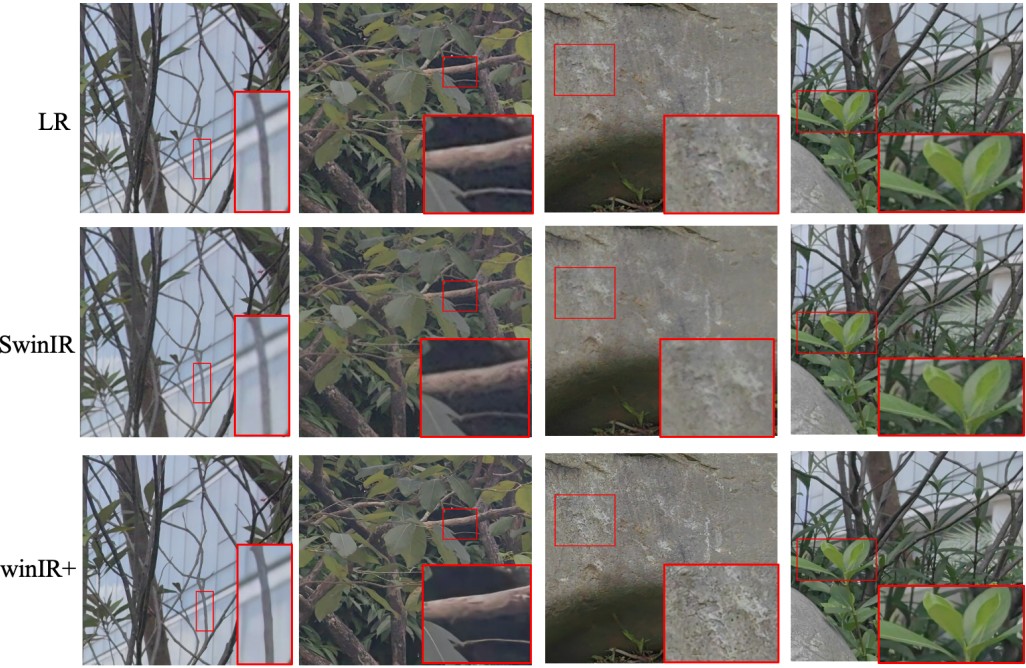

Figure 12: visual comparison of the real images captured by smartphone in the wild.

Here, we present more visual comparisons of the RRDB model on the Mate 50 Pro phone, as shown in Figure 13. It can be observed that our proposed method achieves higher fidelity in image details, closely resembling the ground truth.

LR         RRDB         **RRDB+**         GT

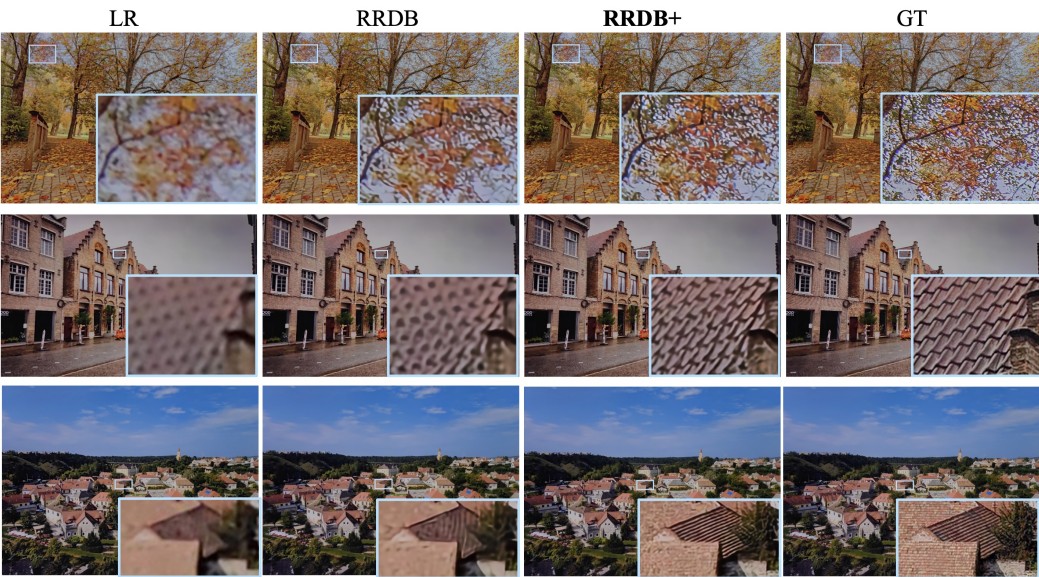

Figure 13: visual comparison of the RRDB model on the Mate 50 Pro phone.

