# OpenReview forum: "Boosting Real-World Super-Resolution with RAW Data: a New Perspective, Dataset and Baseline"
_ICLR.cc/2025/Conference — ICLR 2025 Conference Withdrawn Submission_

### Official Review · Reviewer_9iZA · 2024-10-26

**Soundness:** 2
**Presentation:** 3
**Contribution:** 1
**Rating:** 3
**Confidence:** 4

**Summary:**

This paper constructs a RealSR-RAW dataset containing over 10,000 LR-HR sRGB images along with corresponding LR RAW data, and proposes a RAW adapter to leverage RAW data for improving the performance of previous RealSR models. Additionally, it provides a detailed analysis of why LR RAW data can enhance the performance of RealSR tasks. Experimental results demonstrate the effectiveness of the proposed method.

**Strengths:**

1. The paper is well-written and easy to read.
2. The analysis of detail loss during the ISP process is comprehensive.

**Weaknesses:**

About dataset construction:
1. First, the authors collected HR RGB data from Unsplash. As far as I know, the images on Unsplash inherently contain varying degradations, such as JPEG compression, making them unsuitable to serve as GT data.
2. I am confused about the process of obtaining the LR data. Capturing LR RGB and LR RAW data by photographing HR images displayed on a UHD monitor with a smartphone seems reasonable, but the alignment process is highly challenging. The authors, however, did not provide a detailed explanation of their alignment method.

About experiments:
1. All experimental results in the paper are evaluated based on the proposed benchmark, without validating the effectiveness of their dataset on existing public benchmarks.
2. This paper lacks comparisons with existing synthetic degradation methods. For example, the degradation simulation pipeline proposed in RealESRGAN is also similar to the ISP process, and such methods should be included for comparison.

**Questions:**

1. Why not directly capture HR-LR pairs by using different resolution settings on the smartphone?
2. What is the specific alignment process mentioned in the paper for LR RGB -> LR RAW and LR RGB -> HR RGB?

---

### Official Review · Reviewer_GmGd · 2024-11-03

**Soundness:** 2
**Presentation:** 2
**Contribution:** 2
**Rating:** 3
**Confidence:** 4

**Summary:**

This paper explores using RAW data to improve real-world image super-resolution (Real SR). While most Real SR methods focus on RGB data, this approach leverages both low-resolution (LR) RGB and RAW inputs to generate high-resolution (HR) RGB images. The authors present a new dataset, RealSR-RAW, containing 10,000 pairs of LR RGB, HR RGB, and corresponding LR RAW data, and propose a RAW adapter that integrates RAW data into existing SR models. The results demonstrate that RAW data significantly enhances detail recovery and improves Real SR performance across several evaluation metrics.

**Strengths:**

The paper provides extensive experiments, showing significant improvements in metrics, which demonstrates the effectiveness of incorporating RAW data.

**Weaknesses:**

1。 Lack of Novelty:

The concept of leveraging RAW data in super-resolution is not new. Previous works, such as "Towards Real Scene Super-Resolution with RAW Images" and the NTIRE 2024 Challenge on Deep RAW Image Super-Resolution, have already explored similar methods and datasets.

2. Dataset Redundancy:

Many RAW-to-RGB super-resolution datasets already exist, and LR/HR RAW pairs can be converted to RGB using standard ISP. This raises questions about the unique value RealSR-RAW adds to the field.

3. Unclear Innovation in Approach:

The method mainly combines RAW and RGB data through concatenation and denoising in a straightforward manner. The justification for a new dataset and adapter lacks substantial novelty and distinct advantages over existing methods.

**Questions:**

I do not have other questions.

---

### Official Review · Reviewer_atHp · 2024-11-03

**Soundness:** 3
**Presentation:** 2
**Contribution:** 2
**Rating:** 5
**Confidence:** 4

**Summary:**

This paper introduces a novel approach to real-world image super-resolution (Real SR) by incorporating RAW image data alongside traditional RGB inputs, presenting three significant contributions. The first contribution is the creation of RealSR-RAW, a comprehensive dataset containing over 10,000 paired images with LR and HR RGB images and corresponding LR RAW data. The second contribution is the use of RAW data as a detail-rich complement to RGB-only Real SR. The third contribution is the development of a RAW adapter that integrates RAW data into existing Real SR models. Although the proposed method shows competitive results in quantitative experiments, its statements should be further validated and the paper requires further refinement in its writing.

**Strengths:**

1. Clear Motivation: The motivation behind the proposed method is clearly articulated.
2. Extensive analyzing experiments: Present evaluations of the proposed framework such as  real-world test images, model complexity, and cross-lens generalization ability.

**Weaknesses:**

1. Lack of critical methodological details：
1) While the paper's main contribution is the introduction of a new dataset, neither the main text nor the supplementary materials provide specific descriptions of key settings and processes such as focal length selection, diverse scene collection, alignment procedures, and color correction processes during dataset collection and processing.
2) Regarding the RAW adapter design, crucial design details such as feature extractor and k_learn are missing. Furthermore, Figure 4(b) lacks proper letter annotations, making it difficult to understand and not meeting academic illustration standards.

2. Analysis of experimental results requires improvement
1) The paper states that the proposed RAW Adapter could suppress noise when adding the LR RAW data, which requires additional comparative experiments for validation, such as comparing estimated noise levels before and after applying the adapter.
2) The provided visual results lack convincing power: In Figure 12's third column, SwinIR's result appears more blurry than the LR input, which contradicts common understanding. Moreover, only visual results based on RRDB and SwinIR are presented, but these methods are somewhat dated, and visual comparisons with current methods (e.g., ResShift) are missing.
3) It is suggested to offer more quantitative performance comparisons with raw image enhancement methods as mentioned in the related works.
4) The user study was conducted using only 10 randomly selected images for evaluation, which is too small a sample size to ensure the reliability of the user study results.

3. Non-academic writing issues:
1) In the abstract, it should be written as "over 10,000 pairs" or specifically "11,726 pairs" instead of "10,000 pairs".
2) In the captions of Figures 7-9 in the appendix, "right" and "left" are incorrectly reversed.
3) There appears to be a possible typographical error in line 317 for the word "elevation".

**Questions:**

1. In Section 3.2, based on the description in the paper, i assume residual images were obtained through pixel-level difference. However, since the demosaicing process changes the image dimensions (from H/2*W/2*4 RGGB to H*W*3 RGB image), how were the residual images in Figure 9 calculated and obtained? This part may need more detailed description.

---

### Official Review · Reviewer_Q2zu · 2024-11-03

**Soundness:** 2
**Presentation:** 2
**Contribution:** 2
**Rating:** 5
**Confidence:** 2

**Summary:**

This paper addresses the field of Real-world Super-Resolution and introduces an approach that utilizes LR RAW data as a detail supplement to enhance the performance of Real SR models.(1). The paper presents RealSR-RAW, which is a Real SR dataset containing over 10,000 high-quality pairs of LR and HR RGB images along with their corresponding LR RAW data. (2)By seamlessly integrating the RAW adapter into various popular Real SR methods, the detail representation capability of these methods is improved. (3)The paper also discusses an in-depth analysis and extensive visual comparisons of the effectiveness of the RAW adapter, demonstrating its performance improvements and robustness across various scenarios.

**Strengths:**

This paper provides detailed experimental data, demonstrating excellent performance in both quantitative and qualitative analyses.

**Weaknesses:**

1.The theoretical analysis in the paper is relatively limited.

**Questions:**

1.RAW image super-resolution is very common in the super-resolution field. Given that RAW images can be obtained, is it necessary to continue performing super-resolution on RGB images? As the paper mentioned in the first subsection of Section 5.3,  I think the comparative experiments maybe unfair, as the results of RAW domain super-resolution are greatly influenced by the ISP. Could you provide results that exclude the impact of that?
2.The PSNR and SSIM values provided in the paper are relatively low, indicating that the objective metrics for super-resolution are not satisfactory. Could the authors provide some explanations for this?
3. Have comparative experiments been conducted with the current state-of-the-art (SOTA) work?

---

### Note · Authors · 2024-11-13

I have read and agree with the venue's withdrawal policy on behalf of myself and my co-authors.